

# Combined microbiome and metabolomics analysis of Taorong-type baijiu high-temperature Daqu and medium-temperature Daqu

Yanbo Liu[1,2,3], Junyi Wu[1,2,3], Haideng Li[2,4], Wenxi Liu[1,2,3], Zhenke Zhang[5], Suna Han[5], Jianguang Hou[5] and Chunmei Pan[1,2,3]

[1] Zhengzhou Key Laboratory of Liquor Brewing Microbial Technology, Henan University of Animal Husbandry and Economy, Zhengzhou, China
[2] College of Food and Biological Engineering (Liquor College), Henan University of Animal Husbandry and Economy, Zhengzhou, China
[3] Henan Liquor Style Engineering Technology Research Center, Henan University of Animal Husbandry and Economy, Zhengzhou, China
[4] College of Biological Engineering, Henan University of Technology, Zhengzhou, China
[5] Henan Yangshao Distillery Co., Ltd., Mianchi, China

Corresponding author
Chunmei Pan, sige518888@163.com

## ABSTRACT

**Background:** Daqu is an essential starter for baijiu brewing in China. However, the microbial enrichment and metabolic characteristics of Daqu formed at different fermentation temperatures are still unclear.

**Methods:** High-throughput sequencing technology and the non-targeted metabolomics were used to compare the microbial communities and metabolites of Taorong-type high-temperature Daqu and middle-temperature Daqu. In this study, the relationship between microorganisms and metabolites was established.

**Results:** The study found that the composition and metabolites of the microbial community differed due to the difference in Daqu-making temperature. The bacterial diversity of Taorong-type high-temperature Daqu was higher than that of middle-temperature Daqu, while the fungal community diversity of Taorong-type middle-temperature Daqu was higher than that of high temperature Daqu. A total of 1,034 differential metabolites were screened from the two types of Daqu, and 76 metabolites with significant differences were detected ($P < 0.001$ and variable importance in projection (VIP) > 1.15). Tetraacetylethylenediamine is the metabolite with the largest differential fold among the 76 differential metabolites, which can be used as a potential marker metabolite of high-temperature Daqu.

**Conclusion:** This study helps elucidate the microbial assembly mechanisms and functional expression under different processing conditions through a further understanding of the composition and metabolic profile differences of different types of Daqu microflora in Taorong-type baijiu.

## INTRODUCTION

Chinese baijiu, a traditional Chinese distilled liquor, has a long history and is one of the national cultural features of China (*Xie et al., 2020*). Taorong-type baijiu is the thirteenth type of Chinese baijiu aroma. It is an innovative type of baijiu aroma because it integrates strong aroma, sauce aroma, light aroma and sesame aroma (*Liu et al., 2023*). It is called Taorong-type baijiu because it uses pottery many times in the brewing process. It is loved by consumers because of its clean and refreshing aftertaste (*Hou et al., 2016*). Daqu is a saccharifying and fermenting starter in the production of Chinese baijiu. There are two types of Daqu: high-temperature Daqu (the maximum temperature of fermentation is above 60 °C) and medium-temperature Daqu (the maximum temperature of fermentation is between 50 °C to 60 °C (*Deng et al., 2020*). A starter for the production of Taorong-type baijiu (Fig. 1B). The natural inoculation and open fermentation used in Daqu production enables this process to fully capture the microorganisms in the Daqu-making environment (*Li et al., 2023*). The microorganisms disappear and grow in the starter, and form a unique community structure. Then Daqu is transformed and dried by natural accumulated temperature to form a variety of enzymes, bacteria and substances, which have the functions of saccharification, fermentation and aroma making, and play a vital role in liquor production (*Wu et al., 2019*).

Daqu is rich in microorganisms that can produce many enzymes including protease, esterase and a saccharifying enzyme, amylase (*He et al., 2019*). These substances promote the formation of Daqu flavor compounds (*Jin, Zhu & Xu, 2017*; *Liu & Sun, 2018*). Previous research has shown that that bacteria may be the driving force of microbial succession in Daqu-making process (*Du et al., 2019*). Fungi can secrete various enzymes such as amylase, protease, and cellulase (*Wang et al., 2018*), and the community structure and function of these fungi have an important impact on the quality control of baijiu.

In recent years, researchers have studied the microbial populations of different types of Daqu (*Cai et al., 2021*; *Chen, Wu & Xu, 2014*; *Wang et al., 2017b*) and the Daqu fermentation process (*Xie et al., 2020*) and storage period (*Jiang et al., 2021*) through amplicon sequencing. High-throughput sequencing technology is advantageous for its high throughput and high resolution, and because it can effectively reflect the diversity of microbial species (*Margulies et al., 2005*). Studying the metabolic characteristics of Daqu requires metabolomics technology. *Huang et al. (2020)* used shotgun metagenome sequencing and metabonomics to analyze the microbial and metabolic characteristics in the brewing process of light-flavored baijiu, and evaluated the correlation between microorganisms and their potential functions. *Song et al. (2020)* used metabolomics to distinguish Luzhou-flavor baijiu (distilled liquor) from the Sichuan Basin and the Yangtze and Huaihe River basins by liquid-liquid extraction combined with GC×GC-TOFMS (*Yang, Fan & Xu, 2021*). In order to better understand the composition of Daqu metabolites and the factors affecting fermentation performance, the relationship between the physicochemical properties of Daqu and its metabolic characteristics was determined by metabolomics and correlation analysis. *Deng et al. (2021)* used high-throughput sequencing and gas chromatography-mass spectrometry to profile microbial diversity and
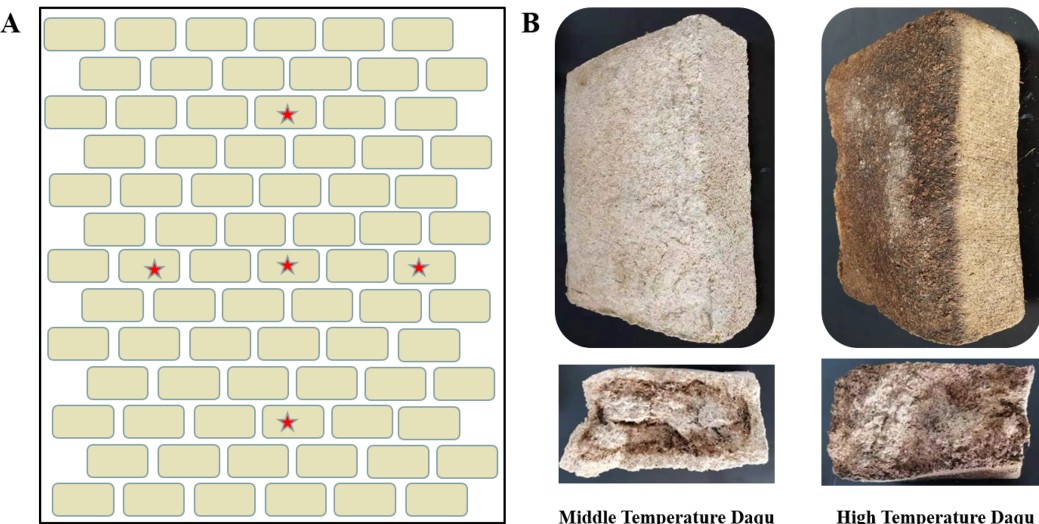

**Figure 1 Schematic diagram of Daqu sampling location (A) and morphological diagrams of medium-temperature Daqu and high-temperature Daqu (B).** The beige rectangle represents the fermented mature Daqu, which is placed layer by layer, and the red pentagram represents the approximate sampling position (A). Daqu volume was about $30 \times 16 \times 12$ cm$^3$ (B).

volatile metabolites during fermentation. *Luo et al. (2022b)* used non-targeted metabolomics to analyze the metabolites of antibacterial black Daqu and black Daqu. They identified the metabolites of the antibacterial black Daqu that inhibit lactic acid bacteria and analyzed the metabolic process of the antibacterial substances. The microbial composition and metabolites of Daqu are important factors affecting the quality of Daqu, and the microorganisms in Daqu not only participate in the main metabolic pathways in the baijiu-making process, but also produce certain flavor components in the baijiu (*Dai et al., 2020*; *Lin, Wu & Xu, 2017*; *Wang et al., 2017a*).

At different fermentation temperatures, the microbiota and metabolites in different starter cultures lead to different characteristics in Taorong-type baijiu products. In recent years, there have been many studies on the microbial community structure of Daqu using microbiomics, and metabolomic methods have been widely used in the study of Daqu metabolites. However, there are only a few previous studies on medium-temperature Daqu and high-temperature Daqu, and the existing studies only focus on the physicochemical properties and the screening and identification of microorganisms. Differences in the metabolic levels of Taorong-type high-temperature Daqu and medium-temperature Daqu, and the relationship between flavor and metabolism have not yet been studied. An analysis of the correlation between endogenous metabolites and flora metabolism in different types of Daqu requires a study of the microbial community structure and its relationship with metabolites in Taorong-type medium-temperature Daqu and high-temperature Daqu.

This study analyzed the microbial composition of Taorong-type medium-temperature Daqu and hightemperature Daqu and used the Daqu metabolic group to comprehensively evaluate the metabolites of the two Daqus. This is the first study to comprehensively analyze the relationship between Taorong-type medium-temperature Daqu and high-temperature Daqu microorganisms and metabolites. The results of this study reveal

the key microorganisms affecting the quality of Taorong-type baijiu and provide a basis for further understanding the material metabolism of Taorong-type baijiu to improve its production and quality and provide clues for the study of other flavored Daqu.

# MATERIALS AND METHODS

## Daqu sample collection

The samples used in this study were collected from Henan Yangshao Distillery Co., Ltd. High-temperature Daqu and medium-temperature Daqu are both prepared by solid-state fermentation from wheat with ingredient formulation, grinding and mixing, shaping, incubation and maturation. The difference in the maximum temperature reached during fermentation distinguishes the high-temperature Daqu (above 60 °C) and medium-temperature Daqu (between 50 °C to 60 °C). Five mature high-temperature Daqu and medium-temperature Daqu (Fig. 1A) were removed from the stack, respectively. Five portions of Daqu in each group were crushed, put into sterile sampling bags, and refrigerated and transported back to the laboratory. Each sample was given a record number and stored at −80 °C. One sample was used for microbial diversity analysis, and another sample was used for UHPLC-QE-MS non-target metabolomics detection. The samples were numbered and grouped as follows: Z1, Z2, Z3, Z4, Z5 (medium temperature Daqu); G1, G2, G3, G4, G5 (high temperature Daqu).

## DNA extraction, PCR amplification, MISEQ library construction

Genomic DNA was extracted using a powerful soil DNA extraction kit (Qiagen, Hilden, Germany) and an Omega genomic DNA extraction kit (Guangzhou Feyou Biotechnology Co., Ltd., Guangzhou, China). DNA quality and concentration were checked using Nanodrop 2000 (Thermo Fisher Scientific, Waltham, MA, USA). According to the specified amplification region, specific primers with barcodes or fusion primers with misplaced bases were synthesized. PCR amplification was performed using 2xTaq Plus Master Mix and corresponding primers. The size of the amplified target band was detected by 1% agarose gel electrophoresis at 170 V for 30 min (Liu et al., 2023).

PCR amplification: The primers were 338F (5′-ACTCCTACGGGAGGCAGCA-G-3′) and 806R (5′-GGACTACHVGGGTWTCTAAT-3′); the primers were SSU0817F (5′-TTAGCATGGAATAATRRAATAGGA-3′) and 1196R (5′-TCTGGACCTGGT-GAGTTTCC-3′).

PCR products were purified automatically using the magnetic bead method. According to the pooling ratio, a certain volume of PCR products was pooled from each sample into a library, and the library fragments were screened using 2% agarose gel. The screened library fragments were detected and quantified by Qubit, 300 ng of library was added to 10 μL end repair and AddA for A tail, and then 33.5 μL adaptor ligation mix was added to connect the sequencing adapter, and the library was purified and recovered. Then, linker primers, enzymes and mix were added for PCR enrichment to complete the library construction. Finally, the constructed Miseq library was purified using the magnetic bead method. The library concentration was checked by Nanodrop, the library fragments were detected by Agilent Bioanalyzer 2100, and the library concentration was accurately quantified by

qPCR (using the StepOnePlus™ real-time fluorescent quantitative PCR system, a dedicated high-throughput sequencing library concentration absolute quantification kit using the Illumina® platform (Yeasen Biotechnology, Shanghai, China) (Li et al., 2016). The final library was sequenced on the Illumina Miseq platform.

## Non-target metabolomic detection by UHPLC-QE-MS

A total of 50 mg of each sample was weighed and 1,000 μL of extraction solution (methanol: acetonitrile: water = 2:2:1 (V/V), containing isotope-labeled internal standard mixture) was added. Each sample was then ground at 35 Hz for 4 min, ultrasonic for 5 min, with this process repeated three times. After grinding, each sample rested at −40 °C for 1 h, then was centrifuged at 4 °C and 12,000 rpm for 15 min, and the supernatant was collected into a sample bottle. The target compounds were chromatographed using a Vanquish (Ultra Performance Liquid Chromatograph from Thermo Fisher Scientific, Waltham, MA, USA) with Waters ACQUITY UPLC BEH Amide 2.1 × 100 mm, 1.7 μm LC column using liquid chromatography. Phase A was an aqueous phase, containing 25 mmol/L ammonium acetate and 25 mmol/L ammonia water, and phase B was acetonitrile. The sample pan temperature was 4 °C and the injection volume was 3 μL. The Thermo QExactive HFX mass spectrometer is capable of primary and secondary mass spectrometry data acquisition (Xcalibur, Thermo, Waltham, MA, USA). The detailed parameters are as follows: Sheath gas flow rate: 30 Arb; Aux gas flow rate: 25 Arb; Capillary temperature: 350 °C; Full MS resolution: 60,000; MS/MS resolution: 7,500; Collision energy: 10/30/60 in NCEmode; SprayVoltage: 3.6 kV (positive) or −3.2 kV (negative).

## Data processing and bioinformatics analysis

After the metabolic raw data was converted into mz XML format by Proteo Wizard software, the self-written R package (the kernel is XCMS) was used for peak identification, peak extraction, and peak alignment and integration. The data was then matched with the KEGG database, human metabolome database, Metlin and CAS for substance annotation, and the cutoff value of the algorithm scoring was set to 0.3. Based on the Spearman correlation analysis method, the correlation between the flora and metabolites was displayed in the form of a correlation coefficient matrix heatmap, using R software 3.3.1, and the R package pheatmap version 1.0.10.

# RESULTS AND ANALYSIS

## Basic sequencing data and alpha diversity analysis

After sequencing the samples, chimera filtering was performed to obtain valid data (clean reads). A total of 674,452 high-quality sequences were obtained from the 16S rDNA sequencing results, with an average of 67,445 per sample; a total of 1,046,622 high-quality sequences were obtained from the ITS (initial transcribed site) results, with an average of 104,662 sequences per sample, and the sequencing coverage was 100%. Non-repetitive sequences were extracted from the effective sequences, and OTU (operational taxonomic unit) clustering was performed on the non-repetitive sequences (excluding single sequences) at 97% similarity. The OTU representative sequences of 411 and 223 bacteria

**Table 1 Sequencing data processing statistics.**

| Sample name | Valid sequence | | OTU number | | Coverage |
|---|---|---|---|---|---|
| | 16S | ITS | OTU number | OTU number | |
| G1 | 76,329 | 69,844 | 285 | 98 | 1 |
| G2 | 51,274 | 70,106 | 200 | 237 | 1 |
| G3 | 49,627 | 114,308 | 213 | 185 | 1 |
| G4 | 102,396 | 135,426 | 182 | 360 | 1 |
| G5 | 124,208 | 71,627 | 183 | 215 | 1 |
| Z1 | 69,212 | 49,309 | 104 | 345 | 1 |
| Z2 | 43,286 | 170,132 | 115 | 341 | 1 |
| Z3 | 48,069 | 202,342 | 118 | 49 | 1 |
| Z4 | 46,323 | 111,386 | 134 | 87 | 1 |
| Z5 | 63,728 | 52,142 | 170 | 300 | 1 |

and 621 and 516 fungi were obtained from high-temperature Daqu and medium-temperature Daqu, respectively. The OTU representative sequences and the number of OTUs in each sample are shown in Table 1.

As shown in Figs. 2A and 2B, as sample sequences increased, the dilution curves of bacteria and fungi gradually flattened, indicating the sequencing depth of this experiment covered most of the microorganisms. The sequencing results accurately reflect the microbial community structure and diversity of high-temperature Daqu and medium-temperature Daqu, indicating that the amount of sequencing data was reasonable. As shown in Fig. 2C, the bacterial Chao1 index of high-temperature Daqu (255.40 ± 37.47) was higher than that of medium-temperature Daqu (134.16 ± 16.55), and the difference was extremely significant ($P \leq 0.001$), indicating that high-temperature Daqu had more bacterial species than medium-temperature Daqu. The bacteria observed_species and PD whole_tree of high-temperature Daqu were higher than those of medium-temperature Daqu. The bacterial Shannon index of high-temperature Daqu (3.90 ± 0.13) was also higher than that of medium-temperature Daqu (3.08 ± 0.29), and the difference was very significant ($P \leq 0.01$). This indicates that the bacterial community diversity of high-temperature Daqu was higher than that of medium-temperature Daqu. An alpha diversity analysis of fungi was performed on Daqu samples, as shown in Fig. 2D. The Chao1 index of medium-temperature Daqu (298.13 ± 118.88) was higher than that of high-temperature Daqu (287.89 ± 179.94), but the difference was not significant ($P \geq 0.05$), indicating that there were slightly more fungal species in medium-temperature Daqu than in high-temperature Daqu. There was no significant difference between high- and medium-temperature Daqu for observed_species and PD_whole_tree. The Shannon index of medium-temperature Daqu (2.69 ± 0.17) was higher than that of high-temperature Daqu (1.43 ± 0.36), and the difference was extremely significant ($P \leq 0.01$), indicating that the fungal community diversity of medium-temperature Daqu was higher than that of high-temperature Daqu.

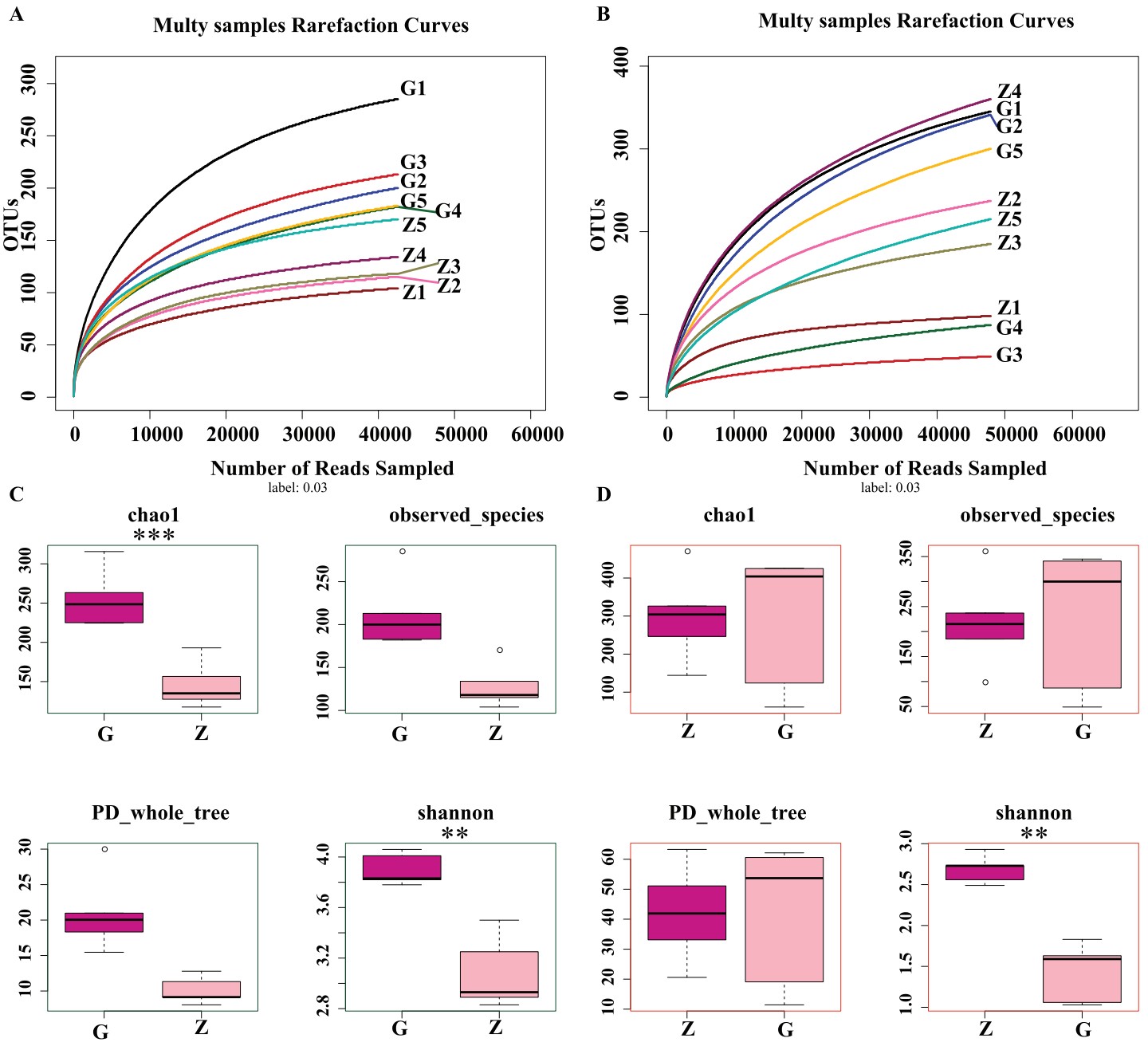

**Figure 2 Daqu bacterial serial dilution curve (A), Daqu fungal dilution curve (B), and Daqu sample bacterial and fungal Alpha diversity analysis box diagram (C and D).** Asterisks (*) represent significant differences, *$P \leq 0.05$, **$P \leq 0.01$, ***$P \leq 0.001$. Chao1: the bacterial species richness index, used to estimate the number of OTUs in the community; observed_species: the number of OTUs actually observed as the sequencing depth increases; PD_whole_tree: pedigree diversity, a diversity index that takes into account species abundance and evolutionary distance; Shannon: one of the indices used to estimate microbial diversity.

In summary, the differences in bacterial community richness between high-temperature Daqu and medium-temperature Daqu were small, but the bacterial community diversity in high-temperature Daqu was much higher than that in medium-temperature Daqu; the difference in fungal community richness between high-temperature Daqu and

medium-temperature Daqu was small, but the fungal community diversity in medium-temperature Daqu was much higher than that in high-temperature Daqu.

## Analysis of differences in microbial community composition in Daqu

The bacteria and fungi of Daqu were classified and analyzed at the phylum level, and the phyla with a relative abundance greater than 1% were defined as the dominant phyla. As shown in Fig. 3A, the four dominant bacterial phyla were Firmicutes, Proteobacteria, Cyanobacteria and Bacteroidota. The four dominant fungal phyla were Ascomycota, Mucoromycota, Basidiomycota and unidentified fungal phyla. As shown in Fig. 3A, the dominant bacterial phyla in high-temperature Daqu were Firmicutes (67.67%), Proteobacteria (21.56%), Cyanobacteria (9.18%) and Bacteroidota (1.04%); the dominant bacterial phyla in middletemperature Daqu were Firmicutes (59.66%), Proteobacteria (33.53%) and Cyanobacteria (6.50%). The dominant fungal phyla in high-temperature Daqu were Ascomycota (97.07%) and Mucoromycota (1.43%); the dominant fungal phylum in medium-temperature Daqu was Ascomycota (97.43%).

The bacteria and fungi of Daqu were then classified and analyzed at the genus level, and the genera with a relative abundance greater than 1% were defined as the dominant genera. Figure 3B shows that the dominant bacterial genera in high-temperature Daqu were *Bacillus* (23.58%), *Erwinia* (18.60%), *Lactobacillus* (16.03%), *unidentified* (9.21%), *Weissella* (8.73%), *Staphylococcus* (7.53%), *Kroppenstedtia* (4.16%), *Pediococcus* (3.67%) and *Leuconostoc* (2.45%); the dominant bacterial genera inmedium-temperature Daqu were *Bacillus* (38.59%), *Cronobacter* (17.58%), *Erwinia* (14.92%), *Lactobacillus* (9.86%), *unidentified* (6.50%), *Staphylococcus* (5.02%), *Leuconostoc* (2.51%) and *Weissella* (1.59%). There were five dominant fungal genera in high-temperature Daqu: *Saccharomycopsis* (72.23%), *Aspergillus* (21.30%), *Rhizopus* (14.10%), *Thermoascus* (1.34%) and *unidentified* (1.24%); there were seven dominant fungal genera in medium-temperature Daqu: *Saccharomycopsis* (38.61%), *Thermoascus* (24.89%), *Aspergillus* (16.52%), *Thermomyces* (24.89%), *Monascus* (5.96%), *unidentified* (2.69%) and *Wickerhamomyces* (1.67%). The effect of temperature on the Daqu bacterial and fungal communities was relatively obvious, and the NMDS model showed that there was also an obvious difference between high-temperature Daqu and medium-temperature Daqu samples (Fig. 3C). The comparison of dominant bacterial genera (Fig. 3D) showed that *Cronobacter* and *Bacillus* were significantly more abundant in medium-temperature Daqu than in high-temperature Daqu, and *Kroppenstedtia*, *Pediococcus* and *Weissella* were significantly more abundant in high-temperature Daqu than in medium-temperature Daqu. The comparison of dominant fungal genera showed that *Saccharomycopsis* was significantly more abundant in high-temperature Daqu than in medium-temperature Daqu, and *Thermoascus* and *Thermomyces* were significantly more abundant in medium-temperature Daqu than in high-temperature Daqu.

## Metabolite difference analysis

It can be seen from Fig. S1 that the baselines of medium-temperature Daqu and high-temperature Daqu were stable, indicating that the instrument data acquisition

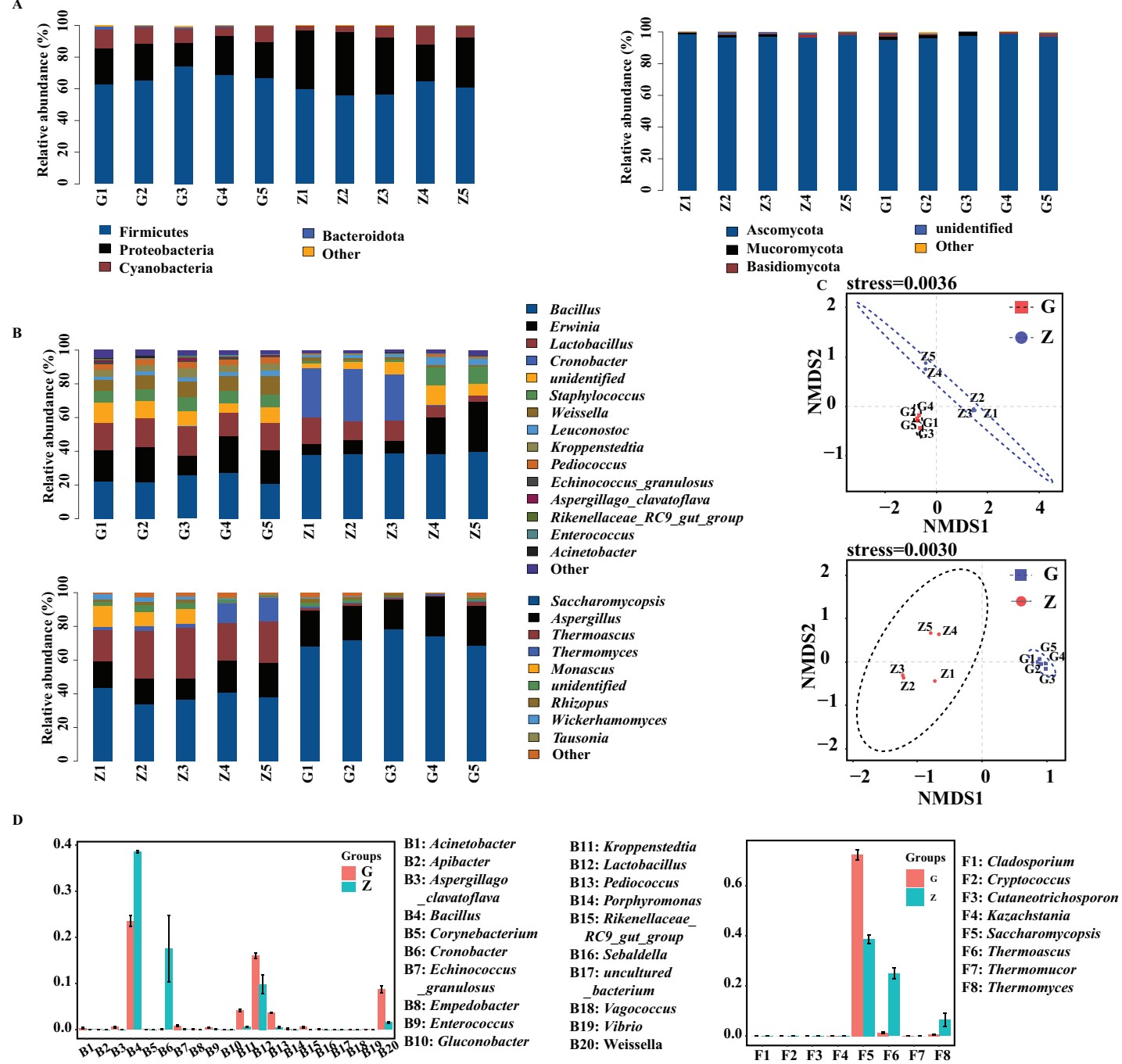

**Figure 3** Bacterial and fungal communities at phylum level (A), genus level (B) in Daqu samples, NMDS graph based on OTUs (C) and genus abundance test graph (D).

stability was very good. The retention time, peak area and number of TIC peaks in positive and negative ion mode were different, so the original data needs to be processed and analyzed in the follow-up.

OPLS-DA was used for better visualization and to further analyze the data. As shown in Fig. 4A, the two groups of samples were clustered into a clearly-distinguished class in the

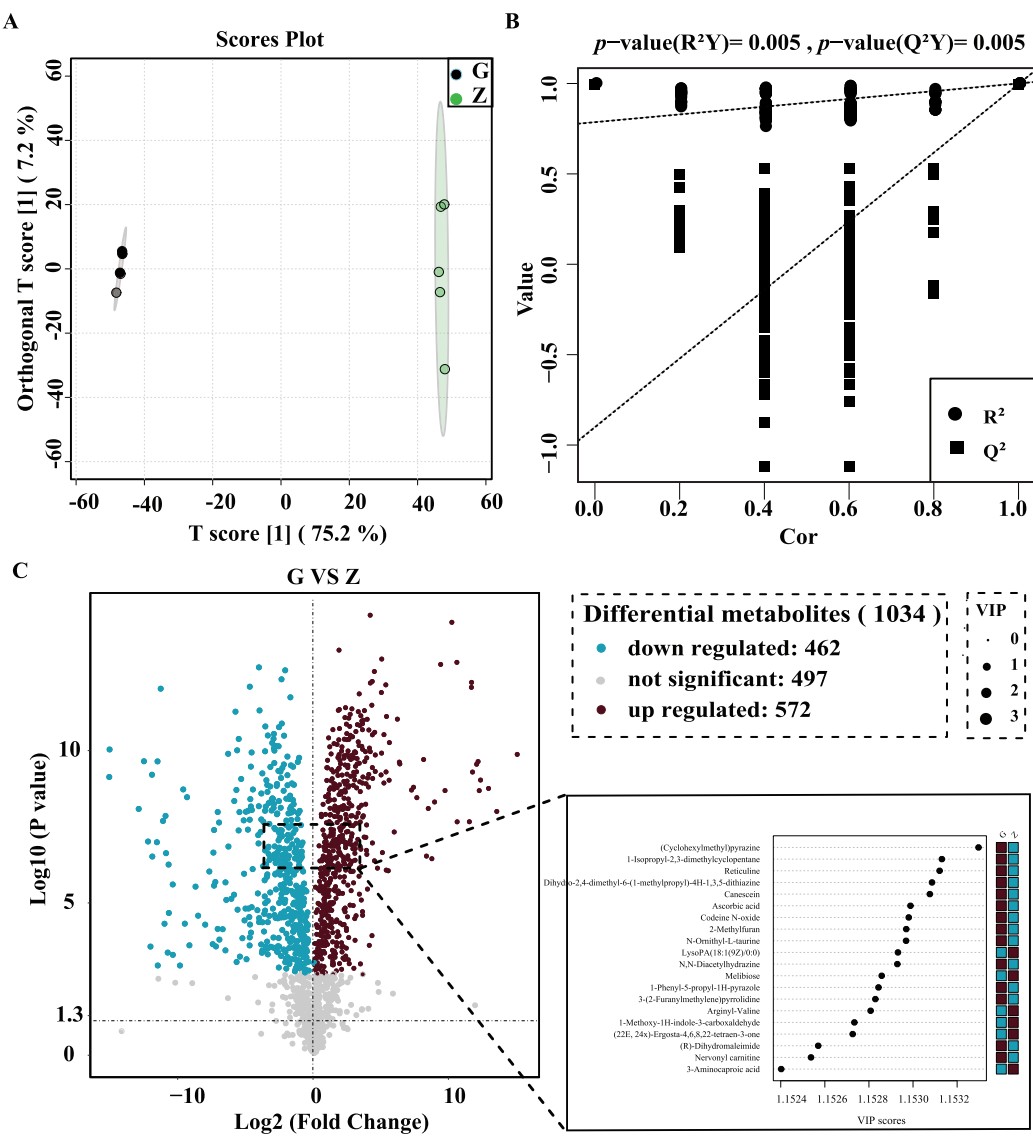

**Figure 4** High-temperature Daqu (red) and middle-temperature Daqu (green) OPLS-DA score (A) permutation test (B) and volcanic map of differential metabolites (C).

interval. The samples were all within the 95% confidence interval. In this model, $R^2$ was the model interpretation rate, $Q^2$ was the predictive ability of the model, and the value of $R^2Y$ was 1, indicating that the established model conformed to the real situation of the sample data, and the original model $Q^2$ was very close to 1 (0.997). High $R^2$ and $Q^2$ values show that the model can explain the difference between the two groups of samples. As shown in Fig. 4B, $R^2Y = 0.025$ and $Q^2 = 0.025$ for the permutation test model, and the Q value of the permutation test random model was smaller than that of the original model, indicating that the original model was robust and there was no overfitting phenomenon. In this study, statistical significance was defined by a $P$ value less than 0.05 and the variable projection importance of the first principal component of the OPLS-DA model was greater than 1.

The differential metabolites were screened and a volcano plot was drawn (Fig. 4C). Compared with medium-temperature Daqu, 1,034 differential metabolites were detected, of which 572 were significantly differentially up-regulated and 462 were significantly differentially down-regulated. In order to find significant metabolites among thousands of variables for further screening, the differential metabolites were further limited to VIP > 1.15, Student's t-test $P < 0.001$, and an absolute value of the logarithm of the difference greater than three. After this screening process, a total of 76 differential metabolites were finally obtained. The specific information of these metabolites is shown in Table 2. These differential metabolites were mainly organic acids and their derivatives, organic heterocyclic compounds, lipids and lipid-like molecules, organic oxygen compounds, phenylpropane and polyketides, benzenes, alkaloids and their derivatives and organic nitrogen compounds. Among the 76 differential metabolites, tetraacetylethylenediamine was the metabolite with the largest difference between medium- and high-temperature Daqu, with a relative content in high-temperature Daqu 37,371 times that in medium-temperature Daqu. This indicates tetraacetylethylenediamine can be used as a potential marker metabolite in high-temperature Daqu.

## Correlation analysis between core microorganisms and differential metabolites

At different fermentation temperatures, microbial groups and metabolites in different fermentation agents may lead to different characteristics of Taorong-type baijiu products, so the influence of Daqu varieties on Taorong-type baijiu cannot be ignored. Different metabolites also had a large influence on the flavor of baijiu. In order to further explore the correlation between microorganisms and metabolites, the correlation between the different metabolites and the core microorganisms of high-temperature Daqu and the core microorganisms of medium-temperature Daqu were analyzed, respectively, and a correlation heat map was obtained.

In high-temperature Daqu, the bacterial genera closely related to the differential metabolites were *Bacillus, Lactobacillus, Weissella, Staphylococcus, Kroppenstedtia*, and the fungal genera with greater correlation with differential metabolites were *Saccharomycopsis*, *Rhizopus*, and *Thermomyces* (Fig. 5A). In medium-temperature Daqu, *Bacillus*, *Cronobacter*, *Lactobacillus*, and *Kroppenstedtia* were the bacterial genera most closely related to differential metabolites, and *Aspergillus*, *Monascus*, *Wickerhamomyces*, and *Rhizopus* were the fungal genera most closely related to differential metabolites (Fig. 5B).

Tyrosol had the highest correlation with *Bacillus* in medium-temperature Daqu, and the highest correlation with *Weissella* and *Thermomyces* in high-temperature Daqu. Xylitol had the highest correlation with *Wickerhamomyces* in medium-temperature Daqu, and the highest correlation with *Weissella* in high-temperature Daqu. Glycine had the highest correlation with *Kroppenstedtia* in medium-temperature Daqu, and had a significant positive correlation with *Staphylococcus* and *Kroppenstedtia* in high-temperature Daqu. Tyrosine had the highest positive correlation with *Cronobacter*, *Lactobacillus* and

Table 2 Differential metabolites of middle-temperature Daqu and high-temperature Daqu.

| Serial number | Metabolites name | G Relative content | Z Relative content | VIP | P | Different value | Differential multiples pair value |
|---|---|---|---|---|---|---|---|
| 1 | D-Xylitol | 1.66E−05 | 0.512830 | 1.151 | 9.09E−11 | 3.23E−05 | −14.92 |
| 2 | Tyrosol | 1.66E−05 | 0.044069 | 1.151 | 2.24E−10 | 0.000376 | −11.38 |
| 3 | Arginyl-Valine | 1.66E−05 | 0.037437 | 1.153 | 9.18E−13 | 0.000442 | −11.14 |
| 4 | "PC(18:4(6Z,9Z,12Z,15Z)/18:1(11Z))" | 0.004051 | 0.290012 | 1.151 | 1.26E−10 | 0.013970 | −6.16 |
| 5 | 5-Hydroxymethyluracil | 0.162112 | 7.968970 | 1.152 | 5.21E−12 | 0.020343 | −5.62 |
| 6 | Oleoyl glycine | 0.007723 | 0.348431 | 1.151 | 9.68E−11 | 0.022164 | −5.50 |
| 7 | Nandrolone | 0.003931 | 0.163684 | 1.151 | 4.09E−11 | 0.024016 | −5.38 |
| 8 | 3-Aminocaproic acid | 0.046583 | 1.100317 | 1.152 | 7.89E−13 | 0.042336 | −4.56 |
| 9 | Proline betaine | 0.070114 | 1.510570 | 1.150 | 1.14E−10 | 0.046416 | −4.43 |
| 10 | LysoPA(18:1(9Z)/0:0) | 0.006300 | 0.093232 | 1.153 | 1.83E−13 | 0.067571 | −3.89 |
| 11 | 3-(1-Pyrrolidinyl)-2-butanone | 0.034893 | 0.500529 | 1.153 | 4.75E−12 | 0.069711 | −3.84 |
| 12 | Isosojagol | 0.114200 | 1.232589 | 1.151 | 1.04E−10 | 0.092651 | −3.43 |
| 13 | Citranaxanthin | 0.003257 | 0.034757 | 1.151 | 1.59E−11 | 0.093717 | −3.42 |
| 14 | o-Tyrosine | 0.076743 | 0.720413 | 1.152 | 3.15E−11 | 0.106530 | −3.23 |
| 15 | Oxoglutaric acid | 0.041461 | 0.341529 | 1.150 | 3.41E−10 | 0.1214 | −3.04 |
| 16 | "3-Ethylidenehexahydropyrrolo[1,2-a]pyrazine-1,4-dione" | 0.305953 | 0.037471 | 1.152 | 3.48E−11 | 8.1651 | 3.03 |
| 17 | "(8S,8S)-Secoisolariciresinol 9-xyloside" | 2.135925 | 0.251989 | 1.151 | 1.72E−10 | 8.4762 | 3.08 |
| 18 | 3-Methyldioxyindole | 0.128723 | 0.014921 | 1.150 | 1.67E−10 | 8.6267 | 3.11 |
| 19 | 1-(4-Methoxyphenyl)-2-nitroethylene | 0.099607 | 0.011504 | 1.151 | 5.10E−10 | 8.6585 | 3.11 |
| 20 | Piplartine | 0.258292 | 0.029242 | 1.150 | 6.57E−10 | 8.8329 | 3.14 |
| 21 | Gravelliferone | 0.181719 | 0.020280 | 1.151 | 1.12E−10 | 8.9605 | 3.16 |
| 22 | Bicine | 0.063014 | 0.006846 | 1.151 | 1.08E−10 | 9.2039 | 3.20 |
| 23 | "2-(1,2,3,4-Tetrahydroxybutyl)-6-(2,3,4-trihydroxybutyl)pyrazine" | 0.045710 | 0.004815 | 1.151 | 3.25E−10 | 9.4931 | 3.25 |
| 24 | 8-Methyldihydrochelerythrine | 0.139397 | 0.014616 | 1.151 | 1.16E−10 | 9.5375 | 3.25 |
| 25 | L-Acetylcarnitine | 0.104866 | 0.010795 | 1.152 | 2.35E−11 | 9.7141 | 3.28 |
| 26 | "2,4-Undecadiene-8,10-diynoic acid 2,3-dehydropiperidide" | 0.359559 | 0.035063 | 1.151 | 1.80E−10 | 10.255 | 3.36 |
| 27 | "1,2,3,4-Tetrahydro-beta-carboline" | 0.231523 | 0.022386 | 1.151 | 2.73E−10 | 10.342 | 3.37 |
| 28 | Nigakinone | 1.580440 | 0.150715 | 1.152 | 1.39E−11 | 10.486 | 3.39 |
| 29 | 2-Furoylglycine | 0.697191 | 0.063976 | 1.152 | 5.09E−11 | 10.898 | 3.45 |
| 30 | 5-Aminoimidazole ribonucleotide | 0.920460 | 0.083978 | 1.153 | 5.21E−12 | 10.961 | 3.45 |
| 31 | 3-(2-Furanylmethylene)pyrrolidine | 1.104129 | 0.096574 | 1.151 | 6.79E−11 | 11.433 | 3.52 |
| 32 | Ethyl beta-D-glucopyranoside | 2.769660 | 0.241547 | 1.153 | 1.36E−12 | 11.466 | 3.52 |
| 33 | "2-Hydroxy-4,7-dimethoxy-2H-1,4-benzoxazin-3(4H)-one" | 0.258502 | 0.022381 | 1.150 | 2.58E−10 | 11.55 | 3.5 |
| 34 | 2-Acetyloxazole | 0.180952 | 0.015256 | 1.152 | 1.48E−11 | 11.861 | 3.57 |
| 35 | N-Acetylarylamine | 1.073279 | 0.086231 | 1.152 | 7.10E−12 | 12.447 | 3.64 |
| 36 | 3-Oxo-carbofuran | 0.254079 | 0.019446 | 1.152 | 7.15E−11 | 13.066 | 3.71 |
| 37 | Oxytetracycline | 0.248746 | 0.018934 | 1.152 | 1.06E−11 | 13.138 | 3.72 |
| 38 | S-Japonin | 0.035190 | 0.002603 | 1.151 | 2.03E−11 | 13.517 | 3.76 |
| 39 | (Z)-3-(1-Formyl-1-propenyl)pentanedioic acid | 0.206567 | 0.015226 | 1.152 | 1.29E−11 | 13.567 | 3.76 |

| Table 2 (continued) | | | | | | | |
|---|---|---|---|---|---|---|---|
| Serial number | Metabolites name | G Relative content | Z Relative content | VIP | *P* | Different value | Differential multiples pair value |
| 40 | Phenylglyoxylic acid | 0.087212 | 0.006393 | 1.151 | 8.97E−11 | 13.641 | 3.77 |
| 41 | Niazirinin | 0.976845 | 0.068884 | 1.152 | 9.96E−12 | 14.181 | 3.83 |
| 42 | L-Tyrosine | 0.254254 | 0.017004 | 1.153 | 4.81E−12 | 14.953 | 3.90 |
| 43 | Isoeugenitol | 0.187519 | 0.012325 | 1.151 | 2.25E−10 | 15.214 | 3.93 |
| 44 | Phaseollidin | 0.077935 | 0.004941 | 1.151 | 1.07E−10 | 15.774 | 3.98 |
| 45 | Anserine | 0.029462 | 0.001749 | 1.151 | 1.71E−10 | 16.848 | 4.07 |
| 46 | N2-(gamma-Glutamyl)-4-carboxyphenylhydrazine | 0.224782 | 0.013035 | 1.152 | 9.27E−12 | 17.245 | 4.11 |
| 47 | Prolylhydroxyproline | 0.283938 | 0.015047 | 1.152 | 2.09E−11 | 18.871 | 4.24 |
| 48 | (Cyclohexylmethyl)pyrazine | 0.059010 | 0.003053 | 1.151 | 8.15E−11 | 19.328 | 4.27 |
| 49 | 1-Phenyl-5-propyl-1H-pyrazole | 1.113191 | 0.055318 | 1.153 | 3.54E−15 | 20.124 | 4.33 |
| 50 | Aldosine | 1.173876 | 0.057873 | 1.153 | 5.96E−13 | 20.284 | 4.34 |
| 51 | 2-Hydroxy-3-methyl-9H-carbazole | 0.094446 | 0.004653 | 1.151 | 1.07E−10 | 20.3 | 4.34 |
| 52 | Reticuline | 0.141346 | 0.006457 | 1.153 | 3.20E−12 | 21.89 | 4.45 |
| 53 | L-2-Aminoethyl seryl phosphate | 0.593441 | 0.026579 | 1.153 | 5.30E−13 | 22.328 | 4.48 |
| 54 | Ascorbic acid | 0.570560 | 0.025241 | 1.151 | 3.00E−11 | 22.604 | 4.50 |
| 55 | Nigellimine N-oxide | 4.134704 | 0.166116 | 1.153 | 2.50E−13 | 24.891 | 4.64 |
| 56 | 2-Hydroxy-4-oxopentanoic acid | 0.506355 | 0.020021 | 1.152 | 2.47E−11 | 25.291 | 4.66 |
| 57 | (R)-Dihydromaleimide | 0.025485 | 0.000975 | 1.150 | 1.37E−10 | 26.137 | 4.71 |
| 58 | N-Carbamoyl-2-amino-2-(4-hydroxyphenyl) acetic acid | 2.184039 | 0.061293 | 1.153 | 1.22E−12 | 35.633 | 5.16 |
| 59 | "Dihydro-2,4-dimethyl-6-(1-methylpropyl)-4H-1,3,5-dithiazine" | 0.022508 | 0.000625 | 1.151 | 1.19E−10 | 36.012 | 5.17 |
| 60 | Homocysteine | 0.203118 | 0.005604 | 1.153 | 9.76E−14 | 36.248 | 5.18 |
| 61 | N-Ornithyl-L-taurine | 0.066312 | 0.001629 | 1.151 | 2.96E−11 | 40.703 | 5.35 |
| 62 | N-Acetyldopamine | 1.366515 | 0.030394 | 1.153 | 6.88E−13 | 44.96 | 5.49 |
| 63 | Nornantenine | 4.950992 | 0.106711 | 1.152 | 7.47E−12 | 46.396 | 5.54 |
| 64 | Erinapyrone C | 0.053203 | 0.000990 | 1.151 | 5.41E−11 | 53.728 | 5.75 |
| 65 | "(1xi,3xi)-1,2,3,4-Tetrahydro-1-methyl-beta-carboline-3-carboxylic acid" | 0.449989 | 0.005535 | 1.152 | 9.19E−12 | 81.304 | 6.35 |
| 66 | Junosine | 0.032936 | 0.000368 | 1.150 | 2.36E−10 | 89.398 | 6.48 |
| 67 | Resveratrol | 0.032953 | 0.000359 | 1.150 | 2.71E−10 | 91.813 | 6.52 |
| 68 | Codeine N-oxide | 0.041471 | 0.000134 | 1.151 | 1.94E−10 | 309.37 | 8.27 |
| 69 | Canescein | 0.021731 | 1.65E−05 | 1.153 | 6.07E−15 | 1,314.7 | 10.36 |
| 70 | "1-Isopropyl-2,3-dimethylcyclopentane" | 0.195037 | 0.000114 | 1.153 | 1.26E−13 | 1,709.2 | 10.74 |
| 71 | "N,N-Diacetylhydrazine" | 0.059486 | 1.65E−05 | 1.153 | 8.33E−13 | 3,598.8 | 11.81 |
| 72 | Nervonyl carnitine | 0.059909 | 1.65E−05 | 1.153 | 5.70E−13 | 3,624.3 | 11.82 |
| 73 | Glycine | 0.062624 | 1.65E−05 | 1.150 | 5.00E−10 | 3,788.6 | 11.89 |
| 74 | "2,3-Dihydro-5-methylthiophene" | 0.077362 | 1.65E−05 | 1.150 | 2.78E−10 | 4,680.2 | 12.19 |
| 75 | Tyrosyl-Lysine | 0.083247 | 1.65E−05 | 1.150 | 2.26E−10 | 5,036.3 | 12.30 |
| 76 | Tetraacetylethylenediamine | 0.617722 | 1.65E−05 | 1.151 | 1.32E−10 | 37,371 | 15.19 |

**Note:**
High-temperature Qu was used as the experimental group, and middle-temperature Daqu was used as the control group. The positive value of the logarithm of the difference indicates that the relative content of metabolites in the high-temperature Daqu was up-regulated, and the negative value of the log-difference indicated that the relative content of the metabolites in the high-temperature Daqu was down-regulated.

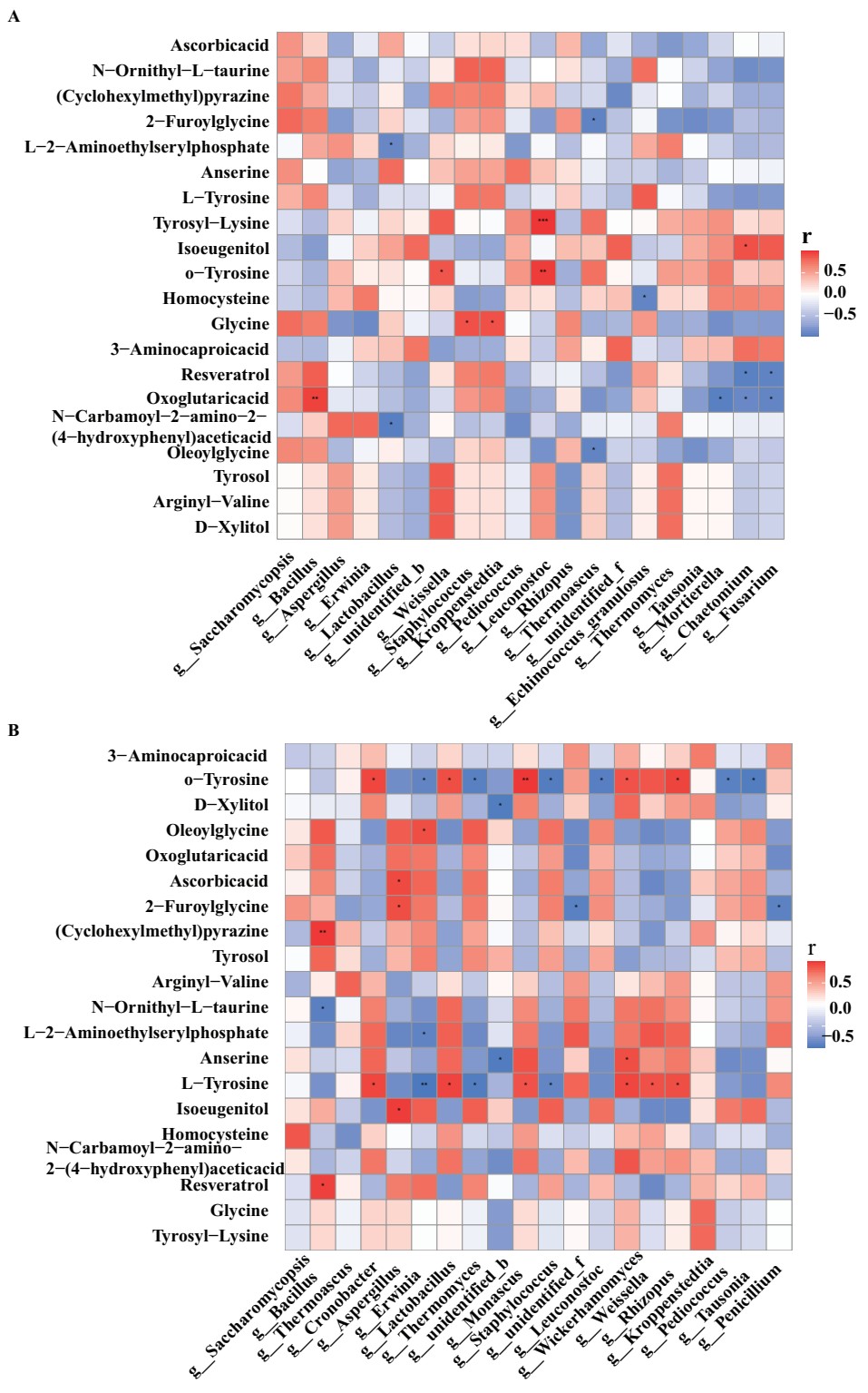

**Figure 5 Heat map of correlations between core microorganisms and metabolites in high-temperature Daqu (A) and middle-temperature Daqu (B).** Red represents positive correlation, blue represents negative correlation, and its color change interval is shown in the diagram on the right side of the figure. The correlation was tested for significance, representing significant differences, $^*P \leq 0.05$, $^{**}P \leq 0.01$, $^{***}P \leq 0.001$.

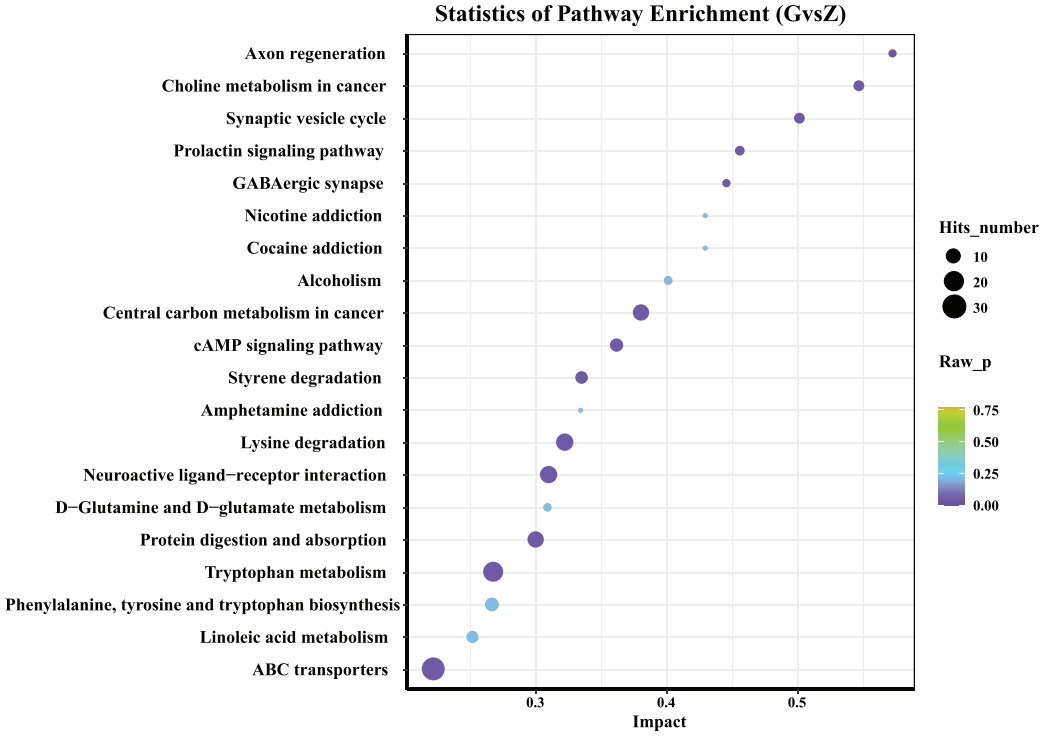

**Figure 6 KEGG analysis of differential metabolites.** The abscissa in the figure represents the impact of each pathway, and the ordinate represents the pathway name. The results of metabolic pathway analysis are presented in bubble charts. The color of the bubble represents the *P*-value of the enrichment analysis. The redder the color, the more significant the degree of enrichment. The size of the dot represents the number of differential metabolites enriched in the pathway.

*Wickerhamomyces* in medium-temperature Daqu, and had a significant negative correlation with *Erwinia*, *Thermoascus* and *Staphylococcus*. Vitamin C had the highest positive correlation with *Aspergillus* in medium-temperature Daqu, and had the highest correlation with *Saccharomycopsis* in high-temperature Daqu.

## Differential metabolite KEGG pathway enrichment analysis

The 20 most significant key metabolic pathways were identified using a KEGG pathway enrichment analysis, according to both *P*-value and impact value. As shown in Fig. 6, in both high-temperature Daqu and medium-temperature Daqu, the differential metabolites were mainly concentrated in the metabolic pathways such as the ATP-binding cassette (ABC) transporters, linoleic acid metabolism, phenylalanine, tyrosine and tryptophan biosynthesis, tryptophan metabolism, protein digestion and absorption, D-Glutamine and D-glucose metabolism.

## DISCUSSION

The microbiome analysis performed in this study showed that the Proteobacteria of medium-temperature Daqu were significantly higher than that of high-temperature Daqu, and the Firmicutes of high-temperature Daqu were significantly higher than that of

medium-temperature Daqu. This may be because Firmicutes have strong heat tolerance, and higher ambient temperature increases their abundance and promotes growth, while heat is not conducive to the growth and reproduction of Proteobacteria. *Chen et al. (2021)* reported that Firmicutes and Ascomycota are the dominant microorganisms in Taorong-type Daqu, which is consistent with the results of this study. Firmicutes and Ascomycota dominated the microbial community, suggesting that they play an important role in the fermentation of Taorong-type baijiu. Bacteroidota and Mucoromycota were the unique dominant microbial phyla in high-temperature Daqu compared with Taorong-type medium-temperature Daqu.

A comparison of the fungal communities of Taorong-type high-temperature Daqu and medium-temperature Daqu found that *Bacillus* was the dominant bacterial genus of both types of Daqu. *Kroppenstedtia* was the unique dominant bacterial genus of high-temperature Daqu compared with medium-temperature Daqu. The relative content of *Bacillus* was lower in high-temperature Daqu than in medium-temperature Daqu, while the relative content of *Weissella* was higher in hightemperature Daqu than in medium-temperature Daqu. This is not completely consistent with the research of *Feng et al. (2021)*, which is likely due to differences in temperature and the unique Daqu-making process of Taorong-type baijiu. *Bacillus* are bacteria that play an important role in flavor in the fermentation of traditional Chinese baijiu (*Luo et al., 2019*). *Weissella* are lactic acid bacteria that have some acid resistance and are widely distributed in fermented products. In the fermentation process, *Weissella* decompose glucose into lactic acid (*Li, Duan & Li, 2006*). *Weissella* are also important microorganisms in baijiu brewing as they are beneficial to the formation of ethyl lactate and the stability of baijiu quality (*Wu et al., 2019*). Previous studies have found that *Kroppenstedtia* are often detected in high-temperature Daqu (*Hou et al., 2022*), indicating that they can adapt to hightemperature environments and have strong heat resistance. The effect of *Kroppenstedtia* on fermentation is not entirely known and should be the target of future research. *Weissella* have strong adaptability in high temperature environments, but hightemperature environments are not conducive to the growth of *Bacillus*.

*Rhizopus* was the dominant fungal genera unique to high-temperature Daqu, and *Thermomyces*, *Monascus* and *Wickerhamomyces* were the dominant fungal genera unique to medium-temperature Daqu. The relative abundance of *Saccharomycopsis* in high-temperature Daqu was significantly higher than in medium-temperature Daqu, while the relative abundance of *Thermoascus* in high-temperature Daqu was significantly lower than in medium-temperature Daqu. *Li et al. (2019)* and other studies have shown that *Saccharomycopsis* can produce enzymes and cellulose degradation-promoting factors, and is a safe bio-resource fungus. *Thermoascus* is one of the dominant fungi of the lush-flavored medium and high-temperature Daqu. It is an important enzyme sourceproducing bacteria in the baijiumaking environment and can promote the degradation of polysaccharides and proteins, which is beneficial to the production of baijiu and aroma in Daqu (*Guo et al., 2017*). A significant increase in *Thermoascus* abundance improves the saccharification and liquefaction power of Daqu (*Luan et al., 2021*). A higher temperature environment is beneficial to the growth of *Saccharomycopsis*, but

temperatures too high inhibit the growth of *Thermoascus* and *Thermomyces*. The optimal temperature range for fungal growth is 20 °C to 30 °C, and temperatures 50 °C to 60 °C are lethal to fungi. Because Taorong-type high-temperature Daqu reaches temperatures higher than 60 °C, only a few fungi with strong heat tolerance can survive the fermentation process of high-temperature Daqu.

Temperature affects the composition and metabolism of microflora. Tetraacetylethylenediamine is a non-toxic and odorless organic compound that is easily biodegraded in the natural environment to form small molecular compounds such as carbon dioxide, water, ammonia and nitrate (*Bian, Dong & Yu, 2021*). There are currently no studies on the formation mechanism of tetraacetylethylenediamine in Daqu, but the results of this study indicate that it could be an intermediate or product of the Maillard reaction process, and temperature increases may promote the formation of tetraacetylethylenediamine. Compared with medium-temperature Daqu, the relative contents of tyrosol and xylitol in high-temperature Daqu were significantly down-regulated, possibly because tyrosol and xylitol largely decomposed due to higher temperatures and the Maillard reaction during the Daqu-making process. Tyrosol is an aroma substance in baijiu, with a very soft aroma and low threshold, but a long duration of bitterness (*Luo et al., 2020*). Compared with medium-temperature Daqu, the relative contents of glycine and tyrosine in high-temperature Daqu were significantly up-regulated. The relative content of glycine in high-temperature Daqu was 3,788.6 times that of medium-temperature Daqu, and the relative content of tyrosine in high-temperature Daqu was 15.214 times that of medium-temperature Daqu. Higher Daqu-making temperatures are conducive to glycine and tyrosine formation. Two amino acids and some sugars may undergo Maillard reactions and produce special fragrance substances. Glycine is an amino acid with a sweet-dominant taste, while tyrosine is an amino acid with a unique umami taste (*Liang et al., 2020*), and glycine and tyrosinecontent help determine the taste of baijiu. Therefore, temperature is one of the most important factors affecting the production of differential metabolites in high- and medium-temperature Daqu.

In previous reports, *Kroppenstedtia* was found in a variety of Daqu (*Li et al., 2019*), but the specific role of *Kroppenstedtia* was not clear. This study was the first to find that *Kroppenstedtia* was positively correlated with glycine, indicating that glycine may be produced by *Kroppenstedtia*. This study was also the first to find that *Bacillus* may be positively correlated with tyrosol and glycine. *Liang et al. (2020)* found that *Bacillus* is one of the key microorganisms of bitter amino acids, affecting the bitter taste of baijiu. Some *Bacillus* species have been reported to break down starch and proteins to produce serine alkaline proteases, metalloproteases, and serine proteases, which contribute to the production of aromatic substances and free amino acids. *Bacillus* also produces acetate and lactic acid, and lactic acid is an important substrate for esterification into ethyl lactate, which is the main aromatic compound in fragrant fermented baijiu. The research of *Hu et al. (2020)* shows that *Weissella* is the main bacterial species of Qing Mao-flavored baijiu. During the fermentation process of baijiu, *Weissella* can convert the sugars in the raw materials into lactic acid and other organic acids, and then react with ethanol to form ethyl lactate and ethyl acetate, which are important to the production of baijiu's mellow and

fruity taste. *Weissella* has a significant role (*Li et al., 2017*), converting fibrous substances into glucose, thereby improving the utilization rate of raw materials, and is the main microorganism in the baijiu making process. *Thermomyces* is the dominant bacterial genus in the fermentation process of Maotai-flavor baijiu (*Zhang et al., 2022a*), and is the source of important enzymes in the baijiu-brewing process. *Wickerhamomyces* is the main aroma-producing yeast strain (*Ming et al., 2015*), mainly used for alcohol production. It can self-metabolize to produce saccharification enzymes for decomposing and utilizing starch and dextrin fermentation, *Wickerhamomyces* also produces a low amount of glycerol, improving the efficiency of alcohol fermentation.

ABC transporters only transport one individual or class of substrates, but its protein family has members that can transport ions, amino acids, nucleotides, polysaccharides, peptides, and even proteins, which is significant to the transport of important nutrients in the lipid bilayer (*Rice, Park & Pinkett, 2014*). During the fermentation process of Daqu and fermented grains, the lipids in the raw materials are metabolized to form linoleic acid, which is an essential fatty acid in human and animal nutrition (*Cai et al., 2019*). In addition, amino acid metabolism is a biological process with a large proportion. Amino acid metabolism produces different metabolites, leading to differences in Daqu appearance and quality (*Zhang et al., 2022b*). The Maillard reaction of amino acids and sugar reduction also influence the Daqu's flavor (*Luo et al., 2022a*).

## CONCLUSION

Through a combined microbiome and metabolomics analysis, this study confirmed the effect of temperature as a key factor on the growth of each type of bacteria in Daqu microflora, thereby affecting the metabolite composition of Daqu. Bacteroidota and Mucoromycota were the unique dominant microbial phyla of Taorong-type high-temperature Daqu, *Kroppenstedtia* was the unique dominant bacterial genus of high-temperature Daqu, *Rhizopus* was the unique dominant fungal genus of high-temperature Daqu, and *Thermomyces*, *Monascus* and *Wickerhamomyces* were the dominant fungal genera that were unique to medium-temperature Daqu. The relative content of tetraacetylethylenediamine in high-temperature Daqu was 37,371 times that of medium-temperature Daqu. Among the 76 differential metabolites identified, tetraacetylethylenediamine was the metabolite with the largest difference between high- and medium-temperature Daqu, indicating that it can be used as a potential marker metabolite in high-temperature Daqu. This study was the first to find that *Kroppenstedtia* was positively correlated with glycine, and that *Bacillus* was positively correlated with tyrosol and glycine. Different Daqu-making temperatures lead to differences in metabolites and microbial community composition. The results of this study lay a theoretical foundation for scientifically guiding the production of Taorong-type baijiu, and add to the understanding of the microbial community structure and metabolism of high-temperature Daqu and medium-temperature Daqu and the impact of metabolites and microorganisms on flavor and quality.

### Funding

This work supported by the Key Technologies Research and Development Program of Henan Province of China (202102110130), the Major Science and Technology Projects of Henan Province of China (181100211400) and the Food Science and Engineering Key Discipline Construction Project of Henan University of Animal Husbandry and Economy (XJXK202203). The Key Research and Development Project of Henan Province (231111112000) and the Key Discipline Construction Project of Food Science and Engineering of Henan University of Animal Husbandry and Economy (XJXK202203) supported the APC for this article. The funders had no role in study design, data collection and analysis, decision to publish, or preparation of the manuscript.

### Grant Disclosures

The following grant information was disclosed by the authors:
Key Technologies Research and Development Program of Henan Province of China: 202102110130.
Major Science and Technology Projects of Henan Province of China: 181100211400.
Henan University of Animal Husbandry and Economy: XJXK202203.
Key Research and Development Project: 231111112000.
Key Discipline Construction Project of Food Science and Engineering: XJXK202203.

### Competing Interests

Zhenke Zhang, Suna Han and Jianguang Hou are employed by Henan Yangshao Distillery Co., Ltd.

### Author Contributions

- Yanbo Liu conceived and designed the experiments, analyzed the data, prepared figures and/or tables, authored or reviewed drafts of the article, and approved the final draft.
- Junyi Wu performed the experiments, analyzed the data, prepared figures and/or tables, and approved the final draft.
- Haideng Li conceived and designed the experiments, performed the experiments, analyzed the data, prepared figures and/or tables, investigated and analyzed the feasibility of the experiment, and approved the final draft.
- Wenxi Liu performed the experiments, analyzed the data, authored or reviewed drafts of the article, and approved the final draft.
- Zhenke Zhang conceived and designed the experiments, authored or reviewed drafts of the article, and approved the final draft.
- Suna Han conceived and designed the experiments, authored or reviewed drafts of the article, used statistical techniques for data analysis and bioinformatics analysis, and approved the final draft.
- Jianguang Hou conceived and designed the experiments, authored or reviewed drafts of the article, and approved the final draft.

- Chunmei Pan conceived and designed the experiments, authored or reviewed drafts of the article, and approved the final draft.

## Data Availability

The raw sequence data are available in the National Center for Biotechnology Information Sequence Read Archive (SRA): PRJNA977569.

## Supplemental Information

Supplemental information for this article can be found online at http://dx.doi.org/10.7717/peerj.16621#supplemental-information.

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
