# Peer review of "Combined microbiome and metabolomics analysis of Taorong-type baijiu high-temperature Daqu and medium-temperature Daqu"

_PeerJ, doi:10.7717/peerj.16621_

## Round 0.1 · original submission · Major Revisions

Please, reviewers have examined your work. Kindly attend to their comments in the very best detail.

Remember to provide detail responses, not just in the revised manuscript, but also as a reply to their comments.

**Language Note:** The review process has identified that the English language must be improved. PeerJ can provide language editing services - please contact us at copyediting@peerj.com for pricing (be sure to provide your manuscript number and title). Alternatively, you should make your own arrangements to improve the language quality and provide details in your response letter. – PeerJ Staff

Reviewer 1 ·

Basic reporting

1. Table 2 lists differential metabolites between both Daqu types. While most entries in this table are clear, some are ambiguous. In some cases, the names used are generic terms, making it difficult to find the specific chemical the authors are referring to. Examples include: #2 - “Mellow”; #13 - “Lemon Yellow”; #14 - “Neighbors”; #15 - “Oxide”; #22 - “Lettuce”; #69 - “Soybean”. In other cases, non-standard notation is used, making it difficult to understand: #39 - “S-Japanese saponin”; #62 - “N-Bird-L-Taurine”; #71 - “N, n-two acetyl”. The chemical nomenclature in this table should be standardized, and if possible - a CAS number should be included to avoid ambiguity.

2. Figure 1A - this schematic is unclear, the meaning of each beige rectangle is not clearly explained. For those not familiar with the Daqu fermentation process, it is not clear if each block is a separate piece of Daqu, or if the entire big rectangle is one piece of Daqu sampled at different points.

3. Figure 1B - please add a scale bar. Those unfamiliar with Daqu fermentation, there is no sense of scale.

4. Section 2.2, line 117: NanoDrop model was not specified, and a comment from the draft was accidentally left in the text.

5. Section 2.2, line 133: The full instrument name is missing (Agilent Bioanalyzer 2100).

6. Section 2.2, line 134: qPCR is mentioned but no method details are given (probe sequence, qPCR instrument details).

7. Several initialisms are not explained the first time they are introduced: ITS (initial transcribed site), OTU (operational taxonomic unit).

Experimental design

1. Section 2.1, sample collection - although 5 separate samples were collected for each Daqu type, they were all collected from a single distillery, which may not be fully representative of this Taorong-style baijiu more broadly.

2. Section 2.1, sample collection - as shown in Figure 1B, there is substantial spatial variation within a cross-section of each Daqu; a sample from the interior may be quite different from a surface sample. The authors describe sampling from 5 locations of each Daqu, but it is not clear if they are only sampling the exterior surface or digging into the internal core of the Daqu.

Validity of the findings

No comments, the conclusions are supported by the data.

·

Basic reporting

The manuscript would benefit from language editing. The method section should be rewritten to use reported speech throughout.

Experimental design

There are major gaps in the description of the protocols, this is a major draw back of the manuscript There is insufficient description of the method to replicate the experiments.

Validity of the findings

All underlying data have been provided; they are robust, statistically sound, & controlled.

Conclusions are well stated and linked to original research question

Additional comments

Abstract
I would suggest that the abstract be properly arranged to flow from background to method, result and conclusion. Separate titled category may not be written but there should be that flow in the abstract
Introduction
What exactly is a Daqu , Taorong and Baijiu? It will be better if these words are fully described when they are first mentioned in the first paragraph of introduction. This helps the international audience of the journal
Lines 51 -52, which results is being referred to here? If it is the ones from this study, then it should not be referenced in introduction section
Line 99 - Recast and check the grammar
Method
How many samples were used?
Line 107-113, section 2.3 – Recast. Method should be written in reported speech format
Section 2.2 should be re-written to sequentially describe the protocol eg
What were the two different DNA extraction kit used?
What were the two pairs of primers listed used for? If two different DNA kits and primers were used, then the corresponding target should be stated
Line 128 – what is the certain amount?
Fungal were stated in the results but the method did not state the procedure for obtaining metagenomics fungi
Others
Line 377 – Used with small u

Reviewer 3 ·

Basic reporting

Comments for the Author
Manuscript Title, Abstract and Introduction
The authors of this study are likely not native anglophones as there are many (too numerous to list) phrases that reflect less comfort in the English language. This concern is not limited to the introduction but rather to the entire manuscript. These issues do not reflect on my review but should be addressed if the manuscript is published. The manuscript should be edited and formatted according to the journal guideline. References appears in different font, font size and are also not consistent.

Experimental design

Insufficient statistical method.

Validity of the findings

Insufficient statistical method, the result was not well discussed but this can improve if the authors revise the manuscript (Please see attached document).

Additional comments

No comments

Annotated reviews are not available for download in order to protect the identity of reviewers who chose to remain anonymous.

---

## Round 0.2 · Minor Revisions

Authors, thank you very much for your patience. Reviewers have favorably considered your revised manuscript. Whilst one is very satisfied with your revisions, the other believes some further actions are required. Please kindly attend to them. Look forward to your revised manuscript. Thank you.

Reviewer 1 ·

Basic reporting

The authors have adequately addressed all reviewer comments. However, the figures and tables may not have been fully updated in the revised version. The authors included corrected versions of Table 1 and Figure 1 in the rebuttal document, but the revised version with tracked changes still has the old version. This may be a simple formatting issue, but I wanted to make sure it is brought to the editor's attention before publication.

Experimental design

The authors have adequately addressed all reviewer comments.

Validity of the findings

The authors have adequately addressed all reviewer comments.

·

Basic reporting

My previous comments have been addressed by the authors.

Experimental design

Please refer to my previous comments

Validity of the findings

Please refer to my previous comments

---

## Round 0.3 · accepted · Accept

I am very satisfied with the revisions authors have made, and the responses provided to the reviewers' comments. The revised manuscript is acceptable for publication. Thank you authors for finding PeerJ as your journal of choice. I look forward to your future scholarly contributions.